# Recognizing the Effect of Ecosystem Disruption on Human Health and Neurodevelopment

**DOI:** 10.3390/ijerph16244908

**Published:** 2019-12-05

**Authors:** Xue Ming, Carly Ray

**Affiliations:** 1Department of Neurology, Rutgers New Jersey Medical School, Newark, NJ 07103, USA; 2Class of 2020, Rutgers New Jersey Medical School, Newark, NJ 07103, USA; cr28@jms.rutgers.edu

**Keywords:** climate change, ecosystem, microbiome, immunologic disease, neurodevelopment

## Abstract

The topics of climate change and ecosystem disruption are at the forefront of global discussion. Accordingly, there is a sense of urgency among citizens, environmental experts, and political leaders for action and policy change. While the effect of a changing climate on the environment is well recognized, its impact on the human body has not been sufficiently described. In our review, we will attempt to outline some of these effects. We will discuss how the recent surge of immunologic disease may be related to the changing profile of microorganisms and antigens in our every-day environment. We will explore how the introduction of antibiotics may result in an altered gut microbiome, and subsequently abnormal neurodevelopment. Finally, we will discuss a possible link between chemical preservatives and neoplastic disease.

## 1. Ecosystem Disruption and Human Health

Climate change and ecosystem disruption are at the forefront of global discussion and debate. A simple browse through the nation’s major news stations will show that there is a clear sense of urgency for action and drastic policy change. From global student rallies to congressional hearings, the concerns of everyday citizens, students, environmental activists, and scientists are being voiced to world leaders. These concerns are shared by medical experts and healthcare professionals. While there has been ample discussion regarding the effect of climate change on the environment surrounding us, we have potentially failed to recognize an effect it may be having on something even more sacred, the human body. I would like to take you on a journey, outlining for you the ways in which ecosystem disruption may be affecting human health, our minds, bodies, and behaviors.

## 2. Ecosystem and Immunology—Allergies and Encephalitis

Deforestation and over-logging have led to a dramatic decrease in the diversity of plant species over the past several decades. As one species of plant becomes extinct, new species emerge to take their place. For example, as oak trees have been excessively harvested for architectural purposes, new species of trees have emerged. With these new trees come new forms of tree pollen, which are inhaled and ingested by humans on a daily basis. Similarly, widespread pesticide use has altered the profile of insects, invertebrates and microorganisms with which we come into contact with through our soil and vegetation.

What is the consequence of this? As the environment is altered by a changing climate, our bodies are bombarded with novel organisms. The molecules which make up these organisms, known in the scientific community as antigens, are recognized as “foreign” by the human body. They elicit a complex inflammatory reaction from our immune system, recruiting specialized cells to eliminate these substances and encode them in their memory should they have to defend against them again. While inflammation was designed as a protective response, frequent and widespread inflammation can become damaging and detrimental, ultimately laying the foundation for chronic disease. Take for example our body’s response to allergens. An allergic reaction occurs when the immune system recognizes a benign substance, such as a peanut, as harmful. Specialized immune cells known as B-cells produce large protein structures called antibodies against this allergen. These antibodies interact with cells throughout the body to trigger release of chemicals such as histamine, which ultimately cause the symptoms of itching, swelling and shortness of breath which are characteristic of an allergic reaction.

As outlined in the Center for Disease Control and Prevention (CDC)’s surveillance and reporting data [1], there has been a shift in the trend of disease over the past several decades, with the prevalence of immunological disorders increasing as the prevalence of infectious disease declines [1,2]. An example of this is the recent surge of allergies which is unlike anything we have ever seen before. Peanut allergy has become one of the most common food allergies in children throughout the US, with many schools responding by becoming “peanut-free”. According to research presented at the 2017 American College of Allergy, Asthma and Immunology (ACAAI) Annual Scientific Meeting, peanut allergy in children has increased 21% since 2010, with nearly 2.5% of children in the U.S. possibly having an allergy to peanuts [3]. As explained by Ruchi Gupta, Professor of Medicine in the Division of Allergy and Immunology at Northwestern University, this surge of allergies is not limited to children. According to a cross-sectional survey study conducted by Gupta and recently published in the January 2019 issue of JAMA Network Open, 10.8% of adults in the US have a food allergy, with 48% of those developing at least one such allergy as an adult [4]. The reason for this rise in allergy remains unknown, however I find it hardly a coincidence that this increase coincides with the extensive shift in antigen profile.

With regard to immunopathology, I (XM) must also mention the uptick in encephalitides I have observed as a practicing neurologist over the past several years. An increasing number of my patients, both children and adults, have presented to me with encephalitis, an inflammation of the brain, of unknown etiology. It has become standard practice to treat such patients with immune suppressants and, in my experience, they typically recover well. However, it is all too common for these patients to suffer a relapse. In my opinion, the etiology of these mysterious encephalitides may be due to the phenomena of molecular mimicry and autoimmunity. When a foreign antigen shares similarity with proteins within our bodies, antibodies created against these antigens can mistakenly attack our own tissue. When this antibody cross-reactivity occurs in the brain, it leads to the devastating neural inflammation we see with encephalitis. In 2015, a case of allergic encephalitis was published in Neurology: Neuroimmunology and Neuroinflammation, describing a wasp sting induced encephalitis with gelastic status epilepticus, confirmed with IgE serology [5]. Unusual causes of encephalitis such as this could become more prevalent in a time when the ecosystem is undergoing dramatic change.

## 3. Ecosystem and Dysbiosis—Effect on Neurodevelopment

The introduction of antibiotics and disinfectants into our ecosystem has resulted in radical changes to the human body. As vectors such as food, soils, lotions and pharmaceutical drugs introduce antimicrobials into our bloodstream, the symbiotic bacteria we rely on for our most essential bodily processes are eliminated. The microorganisms that once maintained a crucial micro-environment in our gut, skin, oronasal cavity and reproductive tract; that allowed for adequate digestion and absorption; have been altered. They are no longer the friends that live within us, but rather a nidus for dysfunction. Microbiome perturbance has become a hot topic in medicine recently. Dysbiosis has been linked with a wide variety of medical pathology ranging from gut regulatory disorders such as small intestinal bacterial overgrowth (SIBO), to cardiovascular and neurologic disease.

As someone who specializes in neurodevelopmental disorders, I have been particularly struck by the increase in autism spectrum disorder (ASD) and attention deficit hyperactivity disorders (ADHD) over the past several decades. According to recent data published by the CDC, the number of children diagnosed with ASD has increased from one in 150 in the year 2000, to one in 59 in 2014 [6]. A similar increase has been seen for ADHD with the number of US children diagnosed with ADHD rising from 4.4 million in 2003 to 6.1 million in 2016 [7]. In addition, depression and anxiety have become common among the pediatric population, with one epidemiological study finding prevalence rates of preschool depression up to 2% [8]. The pathophysiology behind these diseases remains unclear, however, it has been hypothesized that a dysfunctional microbiome may contribute to the pathogenesis. There is a growing body of clinical and bench research supporting the role of microbiota metabolites in central nervous system development and behavior regulation. Several murine models have shown that microbiota deficient mice develop abnormal neuroanatomy and neurotransmitter profiles, and exhibit maladaptive responses to stress [9]. In my own research I have detected abnormal amino acid metabolism, increased oxidative stress, and altered gut microbiomes among a subset of patients with ASD [10].

## 4. Ecosystem and Cancer

With the introduction of preservatives into our natural environment, our daily exposure to toxins has been amplified enormously. The health effects of food preservatives have become a significant source of concern in recent years. In 2015 the World Health Organization’s International Agency for Research on Cancer classified processed meat as carcinogenic to humans based on evidence linking consumption of processed meat to colorectal cancer [11]. A recent meta-analysis showed an association of a 12% increase in the risk for colorectal cancer for each 100 grams per day increase in consumption of red and processed meat [12]. While further research must be done, studies have suggested that nitrates and nitrites, common chemical preservatives added to meat, may be the culprit. In the body these preservatives can lead to the formation of N-nitroso-compounds (NOCs), which suggests a possible association between NOCs and a higher incidence of gastrointestinal cancer, specifically rectal cancer [13]. 

Molecular pathology and epidemiology (MPE) studies suggest that tumor pathogenesis is likely influenced by exogenous and endogenous factors [14]. As such, thorough research on the mechanism of oncogenesis must not only focus on tumor microenvironment but expand to include gut microbial, environmental, nutritional and lifestyle factors. For example, it has been hypothesized that diet and intestinal microbiota, particularly *Fusobacterium nucleatum*, may both be associated with colorectal cancer. An MPE study was conducted to test this association and found that a diet rich in fiber and whole grains was associated with a decreased risk of colorectal cancer in persons containing *F. nucleatum*, suggesting decreased carcinogenesis through suppression of this microorganism. Further MPE studies on pathogenic interactions between the environment, immune system and tumor cells will shed light upon the mechanisms of the effect of ecosystem disruption on human immune system dysfunction and other epigenetic modification of our body, which in turn contributes to tumorigenesis [15,16].

## 5. What Can We Do to Improve the Future of Health?

By now, I hope to have demonstrated how ecosystem disruption has the potential to gravely affect human health. What strategies can we employ to minimize this impact? Should we consider reversing our modern lifestyles? We must familiarize our bodies with the new microorganisms which make up our earth by exposing ourselves to the natural environment more regularly. We must minimize unnecessary antimicrobial exposure. Ultimately, we must halt the destruction of our environment, reduce pollution, save energy, go green, and become conscious about our behaviors, such as by refraining from cigarette smoking. 

## 6. Conclusions

Climate change and ecosystem disruption have the potential to profoundly impact the human body. The call for change, which has been voiced by environmental scientists and political activists, is shared by medical professionals. In this review, we have discussed the relationship between immunologic disease and the changing profile of microorganisms and antigens in our daily environment, between antibiotics and altered gut microbiome with subsequent abnormal neurodevelopment and between exogenous factors such as chemical preservatives and neoplastic disease. Heterogeneity of disease pathogenesis is a core scientific principle, and the effect of a changing environment on human health must be explored. Transdisciplinary fields such as MPE offer an avenue for such research and the opportunity to establish causality. The bottom line is, eco disruption affects human health. This is a world-wide issue that requires immediate attention. To protect the environment is to protect ourselves and our children.

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
