# Peer review of "Recognizing the Effect of Ecosystem Disruption on Human Health and Neurodevelopment"

_ijerph, 2019, doi:10.3390/ijerph16244908_

Round 1
Reviewer 1 Report
This is an interesting article on very hot topics. The paper is generally well written. It covers a good range of topics and is successfully focused on environment, ecosystem, and microbiome. I have some comments for improvement.
Considerable issues are scientific rigor and reproducibility. It is challenging to establish causality. This is in part due to lack of recognizing and addressing suboptimal study designs in studies and inadequate replication efforts within a given study. Mechanistic understanding is also limited. For a better design, adequate expertise in biology, pathology, biostatistics, and epidemiology must be included in such research efforts.
Related to the above point, the author can discuss the recent trend of molecular pathological epidemiology (MPE). MPE is an emerging field that can link environment (including the microbiome) to molecular pathologies (including immunity) using robust epidemiological study designs. It can provide mechanistic understanding and help us understand causal relations. Hence, MPE can contribute to biomarker research and precision medicine. You can easily find relevant references on MPE by net search (eg, Gut 2011; Annu Rev Pathol 2019).
Author Response
We appreciate this reviewer's caution on causal mechanic statements of our cited studies. We have now revised these statements accordingly (in red text).
A recent meta-analysis showed an association of a 12 percent increase in the risk for colorectal cancer for each 100 grams per day increase in consumption of red and processed meat.[12]. While further research must be done, studies have suggested that nitrates and nitrites, common chemical preservatives added to meat may be the culprit. In the body these preservatives can lead to the formation of N-nitroso-compounds (NOC’s), which suggests a possible association between NOC’s and a higher incidence of gastrointestinal cancer, specifically rectal cancer.[13].
We are grateful for the suggestion of Molecular Pathology Epidemiology for rigorous scientific design and the link in the crossroad of the multidisciplines. The following paragraph is added in this revised manuscript.
Molecular Pathology and Epidemiology (MPE) studies suggest that tumor pathogenesis is likely influenced by exogenous and endogenous factors [14]. As such, thorough research on the mechanism of oncogenesis must not only focus on tumor microenvironment but expand to include gut microbial, environmental, nutritional and lifestyle factors. For example, it has been hypothesized that diet and intestinal microbiota, particularly Fusobacterium nucleatum, may both be associated with colorectal cancer. An MPE study was conducted to test this association, and found that a diet rich in fiber and whole grains was associated with a decreased risk of colorectal cancer in persons containing F. nucleatum, suggesting decreased carcinogenesis through suppression of this microorganism. Further MPE studies on pathogenic interactions between the environment, immune system and tumor cells will shed light upon the mechanisms of the effect of ecosystem disruption on human immune system dysfunction and other epigenetic modification of our body, which in turn contributes to tumorigenesis. [15,16]
Reviewer 2 Report
Dear Autjors,
My review is attached as a pdf file.
Reviewer

Author Response
Thank you very much for the enthusiasm. We share your point on cigarette smoking on environment and added into this sentence below.
we must halt the destruction of our environment - reduce pollution, save energy, go green and become conscious about our behaviors such as refrain from cigarette smoking.